# Ultrafast light field tomography for snapshot transient and non-line-of-sight imaging

Xiaohua Feng[1] & Liang Gao [1,2,3] ✉

Cameras with extreme speeds are enabling technologies in both fundamental and applied sciences. However, existing ultrafast cameras are incapable of coping with extended three-dimensional scenes and fall short for non-line-of-sight imaging, which requires a long sequence of time-resolved two-dimensional data. Current non-line-of-sight imagers, therefore, need to perform extensive scanning in the spatial and/or temporal dimension, restricting their use in imaging only static or slowly moving objects. To address these long-standing challenges, we present here ultrafast light field tomography (LIFT), a transient imaging strategy that offers a temporal sequence of over 1000 and enables highly efficient light field acquisition, allowing snapshot acquisition of the complete four-dimensional space and time. With LIFT, we demonstrated three-dimensional imaging of light in flight phenomena with a <10 picoseconds resolution and non-line-of-sight imaging at a 30 Hz video-rate. Furthermore, we showed how LIFT can benefit from deep learning for an improved and accelerated image formation. LIFT may facilitate broad adoption of time-resolved methods in various disciplines.

[1] Department of Bioengineering, University of California, Los Angeles, CA, USA. [2] Department of Electrical and Computer Engineering, University of Illinois at Urbana-Champaign, Urbana, IL, USA. [3] Beckman Institute for Advanced Science and Technology, University of Illinois at Urbana-Champaign, Urbana, IL, USA. ✉email: gaol@ucla.edu

Time-resolved imaging[1,2] plays pivotal roles in a range of scientific studies in biology, chemistry, and physics. Despite of its widespread impact and applications, fast acquisition of large-scale 2D time-resolved data with a picosecond resolution remains a long-standing challenge to solve. To date, streak cameras and intensified charge-coupled device (ICCD) sensors are parallel detectors of choice for measuring ultrafast dynamics. It is, nevertheless, necessary to perform extensive scanning either in the spatial domain (for streak cameras[3,4]) or temporal dimension (for ICCDs[5]) to obtain a 2D time-resolved data, which is an inherently time-consuming process. A single-photon avalanche diode[6] (SPAD), an emerging ultrafast detector with exceptional sensitivity, can achieve a temporal resolution of tens of picoseconds and holds great potential to be fabricated in large-format two-dimensional arrays[7]. However, obtaining a grayscale time-resolved data still requires temporal scanning or repeated illuminations with a time correlated single-photon counter (TCSPC), which leads to an inferior filling factor for 2D SPAD sensors[8] given current fabricating technologies. The need for scanning also undesirably restricts the applicable scope of these cameras to strictly repeatable events.

The past decade has witnessed the development of a plethora of ultrafast cameras capable of 2D time-resolved imaging with a single snapshot. However, none of these methods attained the challenging combination of a deep sequence (of over 1000) and a picosecond temporal resolution, even if active methods are considered. For instance, using specialized illumination, sequentially timed all-optical mapping photography (STAMP)[1] can achieve a temporal resolution of 200 fs, but only a rather limited sequence depth (<50) is obtainable. On the other extreme, serial time-encoded amplified imaging[2] (STEAM) can stream 2D images continuously while its temporal resolution is restricted to a few nanoseconds. Up to now, compressive ultrafast photography (CUP)[9] has been the only passive camera that offers a three-dimensional data cube (x, y, t) over $100 \times 100 \times 100$ in a single snapshot and reached a sub-picosecond resolution[10,11]. Unfortunately, it is challenging to scale it further for larger scale measurements: apart from its inherent trade-off between the spatial resolution and sequence depth, its large compression factor and spatial-temporal cross talk directly limit its achievable spatiotemporal resolution in transient imaging.

The lack of a general tool for single-shot acquisition of large-scale 2D time-resolved data and the inability to cope with extended 3D scenes not only restrict the visualization of transient phenomena in direct view, but also compromise the capability to see around occlusion or, non-line-of-sight (NLOS) imaging. While looking beyond direct view finds broad applications in domains like navigation, surveillance, and even medical imaging[12], current NLOS imagers[12–17] still lag far behind their line-of-sight counterparts in achieving video-rate imaging, though recent work[14] has opened the pathway of systematically transferring line-of-sight imaging methods to the NLOS domain. The major bottleneck, with the computationally intensive reconstruction being lifted off by faster inversion algorithms[13,18,19] and parallel computing[20,21], remains to be the slow acquisition of large-scale time-resolved data. Although edge-resolved transient imaging (ERTI)[22] made a stride to use far fewer scanning for NLOS imaging, it only yields a 2.5D (rather than a full 3D) reconstruction, and its differential measurement still leads to a long exposure time (>10 s) at each scanning position. Faster scanning can also be achieved in several other ways: shortening the sensor exposure time, reducing the spatial scanning density, or parallelizing acquisition[23]. Nevertheless, the scanning mechanism still persists, and the resultant smaller photon counts from shorter exposure typically need to be compensated by using a higher laser power and/or retro-reflective targets[18]. The inability to cope with extended 3D scenes also precludes field-deployable NLOS imaging, which needs to accommodate non-planar or even disconnected surfaces. These obstacles make NLOS imaging arguably one of the most challenging applications for ultrafast cameras.

Here, we present light field tomography (LIFT), an imaging method that is highly efficient in recording light fields and enables snapshot acquisition of large-scale 2D time-resolved data. This is achieved by transforming a one-dimensional (1D) sensor to a 2D light field camera, exploiting the fact conventional light field acquisition is highly redundant—the sub-aperture images are mostly the same except for disparity cues. The vastly faster frame rate of 1D sensors also benefits LIFT for high speed imaging. While prior state-of-the-art ultrafast cameras are severely limited in pixel resolution that prevents light field acquisition, LIFT offers an elegant way to break this restriction. Coupled with a streak camera, LIFT can capture the complete four-dimensional spatiotemporal space in a single snapshot and provide an image resolution over $120 \times 120$ with a sequence depth beyond 1000, enabling unprecedented ultrafast imaging capabilities, including but not limited to, video-rate NLOS imaging using a low laser power.

## Results

**LIFT camera.** The core idea of LIFT is to reformulate photography as a computed tomography (CT) problem[24] by using cylindrical lenses to acquire en-face parallel beam projections of the object. The principle is illustrated in Fig. 1a, showing the transformation of point sources in the object space into parallel lines in the image plane by a cylindrical lens. The line direction in the image space is parallel to the invariant axis—the axis without optical power—of the cylindrical lens. Such an optical transformation of a scene can be artificially decomposed into two steps, as depicted in Fig. 1b: a first step of pinhole image formation and a second step of convolution with a line-shaped point spread function (PSF) that is parallel with the cylindrical lens' invariant axis. The line-shaped PSF allows an individual camera pixel to integrate the image along that line. With a 1D sensor positioned at the center of the image space, a parallel beam projection of the image is acquired along the invariant axis direction. Projection at different angles can be recorded by rotating the cylindrical lenslet with respect to the 1D sensor. By using an array of cylindrical lenslets oriented at distinct angles, one can obtain enough projections simultaneously to recover the image with a single snapshot. Furthermore, because each lenslet observes the same scene from different perspectives, the light field of the scene is naturally sampled in the projection data with an angular resolution equal to the number of lenslets. Such tomographic light field recording is orders of magnitude more efficient than conventional approaches (Methods section). This endows LIFT with full-fledged light field imaging[25,26] capabilities, including depth retrieval, post-capture refocusing, and extended depth of field.

Formally, as analyzed thoroughly in Supplementary Note 1, the light field data acquisition of LIFT can be encapsulated into a single equation: ignoring image magnification, the projected coordinate of a point source located at $(x_0, y_0)$ is $x_l = -x_0 - y_0 \tan\theta + u$ on the 1D sensor, where $\mu$ denotes the angular component contributed by the lenslet array and $\theta$ is the orientation angle of the lenslet. The acquired projection data in LIFT relates to the en-face object via the Fourier slice theorem after computational resampling. The imaging process can be written as (Methods section):

$$b = Ag \qquad (1)$$

where $b$ is the measurement data, $g$ is the vectorized

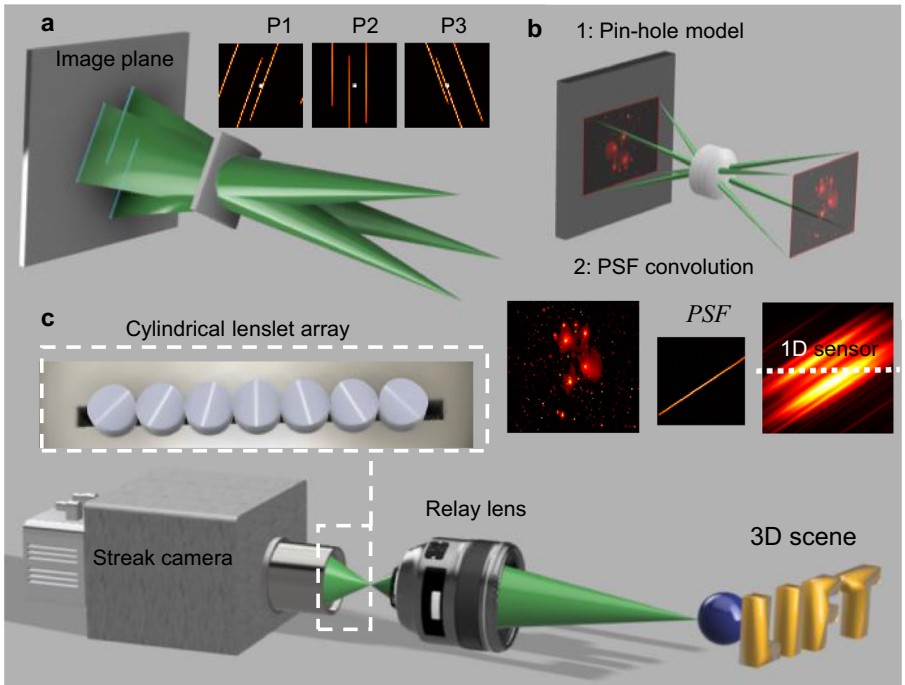

**Fig. 1 Working principle and implementation of light field tomography. a** Illustration of image formation by a cylindrical lens. Three point sources in the object space are transformed into parallel lines on the image plane, producing a projection image. Acquiring such projection images from different perspectives using lenslets oriented at different angles naturally samples the light field of the 3D scene, as exemplified in the insets P1–P3, where the image center is highlighted to visualize the disparities. **b** Two-step modeling of cylindrical lenslet imaging process. For clarity, an image showing predominantly point-like structures is rendered. The 1D projection data is obtained by sampling the convolution result of the pinhole image and line-shaped PSF. Recording such 1D data over time yields a time-resolved measurement. **c** Typical system setup of a LIFT camera. The cylindrical lenslet array is closely secured to the entrance slit of the streak camera.

two-dimensional (2D) image, and **A** is the forward operator representing the parallel beam projections at different angles. The underlying image can be recovered by inverting the above equation with a range of methods such as the analytic filtered backprojection[24]. In theory, one should use as many lenslets as dictated by the Nyquist sampling criterion for high-quality image reconstruction. This is generally impractical for high-resolution 2D imaging due to the limited pixel number of 1D sensors. However, under the framework of compressive sensing, the number of projections required for image reconstruction can be substantially reduced.

A key observation here is that high dimensional data tends to be highly compressible[27]—the spatial image (*x, y*) at each time instance of the spatiotemporal datacube (*x, y, t*) is far simpler than natural photographs and consequently can be efficiently encoded in certain representation bases. Particularly for NLOS imaging, the instantaneous image on the wall can be represented with only ~tens of projections for high quality reconstruction of complex hidden scenes (Supplementary Note 6.2). This particular embodiment renders LIFT similar to sparse view CT[28], which generally requires slower iterative methods for reconstruction and is prone to degraded image quality in certain scenarios (Supplementary Note 2). To mitigate these two issues, we devised a deep adjoint neural network (DANN) to accelerate and improve LIFT image recovery, which incorporates the adjoint operator **A**[T] of the system into a deep convolutional neural network and thereby avoids the blind end-to-end training typical in previous endeavors[29]. This facilitates the deep neural network to generalize well even when it is trained on a small dataset[30]. The synergy between compressive data acquisition and fast deep neural network reconstruction breaks the data bandwidth limit of conventional cameras and enables high-resolution 2D imaging with 1D sensors.

The ultrafast LIFT system configuration is diagrammed in Fig. 1c. Seven customized cylindrical lenslets (diameter, 2 mm; focal length, 8 mm) oriented at distinct angles are assembled on a 3D printed holder and aligned with the slit of a streak camera. The lenslet arrangement—the sequence of the invariant axis' angles with respect to the slit—can be optimized for different applications, such as an automatically extended depth of field (Supplementary Note 3.3). The 3D scene is imaged by a camera lens to the intermediate image space, from which the cylindrical lenslet array forms differently projected sub-images onto the slit plane. A field stop at the intermediate image plane reduces the field of view to avoid the sub-image overlap between the adjacent lenslets. The streak camera relays the 1D projection images from the slit onto a photocathode, converts it to the electronic domain, and eventually deflects it onto different rows of a CCD camera according to the photons' time of arrival. Because the temporal axis is orthogonal to the 1D projection image, there is no spatial-temporal coupling in LIFT, leading to an optimal temporal resolution.

**Three-dimensional transient imaging.** To demonstrate LIFT in ultrafast imaging, we captured a light-in-flight scene that is beyond the capability of existing ultrafast cameras. A light-diffusing fiber[31], whose internal nanostructures scatter out a small fraction of light from its core, was wrapped into a helical shape with a depth range stretching over 80 mm (Fig. 2a). After coupling a picosecond pulsed laser into the fiber, the internal laser pulse evolution was recorded at 0.5 T frames per second with a native temporal resolution of ~3 ps.

Spanning a large depth range, it is challenging for cameras with a fixed focus to well resolve the helical fiber. The common wisdom

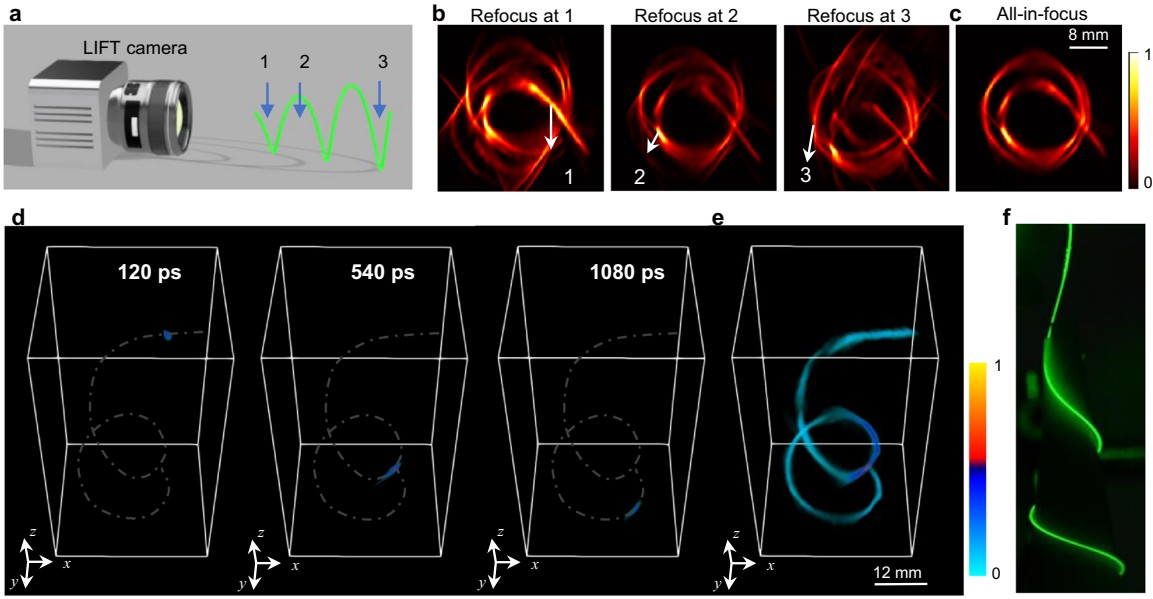

**Fig. 2 Transient light field imaging by a LIFT camera. a** Experimental setup. **b** Post-capture refocusing at different depths. The images are time-integrated so that all parts of the helical fiber can be rendered, which are otherwise separated at different time instances. **c** All-in-focus time-integrated imaging results. **d** 4D (3D space and time) characterization of a picosecond laser pulse propagating inside a light-diffusing fiber. The helical fiber structure is crafted and overlaid in each 3D frame for visual guidance. **e** Time-integrated 3D image of the helical fiber. **f** Photograph of the helical fiber.

is to reduce the camera's aperture for an extended depth of field, which scales poorly for ultrafast imaging owing to the fundamentally limited photon budget at high frame rates. This is illustrated in Fig. 2b, which shows the fiber images obtained at different focal settings, emulated by computationally refocusing the LIFT camera at different depths and integrating time-resolved images along the temporal axis. For each preset focal depth, only part of the helical fiber remains sharp, as indicated by the arrows, and ghost images begin to emerge for heavily defocused parts (Supplementary Note 3.1). In striking contrast, LIFT can synthesize an all-in-focus image (Fig. 2c) to resolve the entire helical fiber structure by leveraging its post-capture refocusing capability (Supplementary Note 3.2). With seven angular components herein, LIFT can effectively increase the depth of field by seven folds, which is notably achieved without compromising light throughput.

Moreover, LIFT enables the extraction of the scene depth at each time instant via the depth-from-focus[32] method, thereby revealing the complete 4D spatiotemporal dimensions of the event under observation. For our current implementation, the depth retrieval accuracy without the relay system is $d^2\Delta L(Da)^{-1} \approx 2$ mm (Supplementary Note 3.4), with $d$ and $a$ being the distance from the lenslet array to the object and 1D sensor, respectively. The lenslet array baseline $D$ and the pixel size $\Delta L$ serve similar roles as those in stereo methods[32]: a large baseline and a smaller pixel yield a better depth resolution. After depth extraction and extending the depth of field, the 3D imaging of laser pulse propagation inside the helical fiber is illustrated in Fig. 2d at several representative time instants, and the complete animation is provided in Supplementary Movie 1. The retrieved 3D structure of the fiber (Fig. 2e), obtained by integrating all frames, agrees qualitatively well with the photograph in Fig. 2f, validating LIFT's capacity in visualizing extended 3D objects. Such extended depth of field and 3D imaging capabilities are defining features of LIFT over other 2D ultrafast cameras[33,34] (Supplementary Note 7).

**Deep adjoint neural network for LIFT reconstruction.** The deep adjoint neural network (Fig. 3a) is critical for accelerating the reconstruction and improving the image quality by learning and

mitigating the system's implementation limitations. Specifically, DANN can alleviate the limited view problem[35], which refers to a degraded tomographic image reconstruction when the projection data does not span the complete angular range of [0°, 180°]. As analyzed in Supplementary Note 2, though not inevitable, a frequency cone along the $k_y$ direction in the $k$-space is not sampled in our current LIFT camera. This is manifested in the all-in-focus image of the helical fiber in Fig. 2c: the horizontal features on the top and bottom parts show an inferior resolution and consequently appear dimmer. However, by training the DANN with a dataset containing similar challenging cases, the network can efficiently learn and mitigate this problem.

To demonstrate our approach, we changed the cylindrical lenslet arrangement for an automatically extended depth of field (dubbed as depth-of-field version in Supplementary Note 3.3) and trained the DANN network for the system using an image set collected from MNIST[36] and FashionMNIST[37] dataset. The training set was created such that ~60% of its images contain rich spatial frequencies inside the missing cone of the system to enable efficient learning of reconstruction under limited view constraints. Figure 3b shows representative DANN reconstructions for LIFT imaging (no temporal deflection) of an experimental test dataset displayed on a monitor. The test dataset was composed primarily with images showing strong features along the horizontal direction to illustrate the pessimistic recovery performance for the scenes afflicted by the limited view problem. While iterative results tend to blur horizontal features as for the helical fiber, the DANN network clearly recovers the images with most horizontal features well delineated. More test dataset comparisons are provided in Supplementary Note 8.

The laser pulse propagation inside the helical fiber is re-captured using the automatically extended depth-of-field version of LIFT but re-wrapped to emphasize its horizontal features for an escalated limited view problem. The recovered images at representative time instants by iterative methods and DANN are compared in the first and second row of Fig. 3c (full clips in Supplementary Movie 2). As the laser pulse propagated to the horizontal parts (top and bottom), iterative results get dimmer

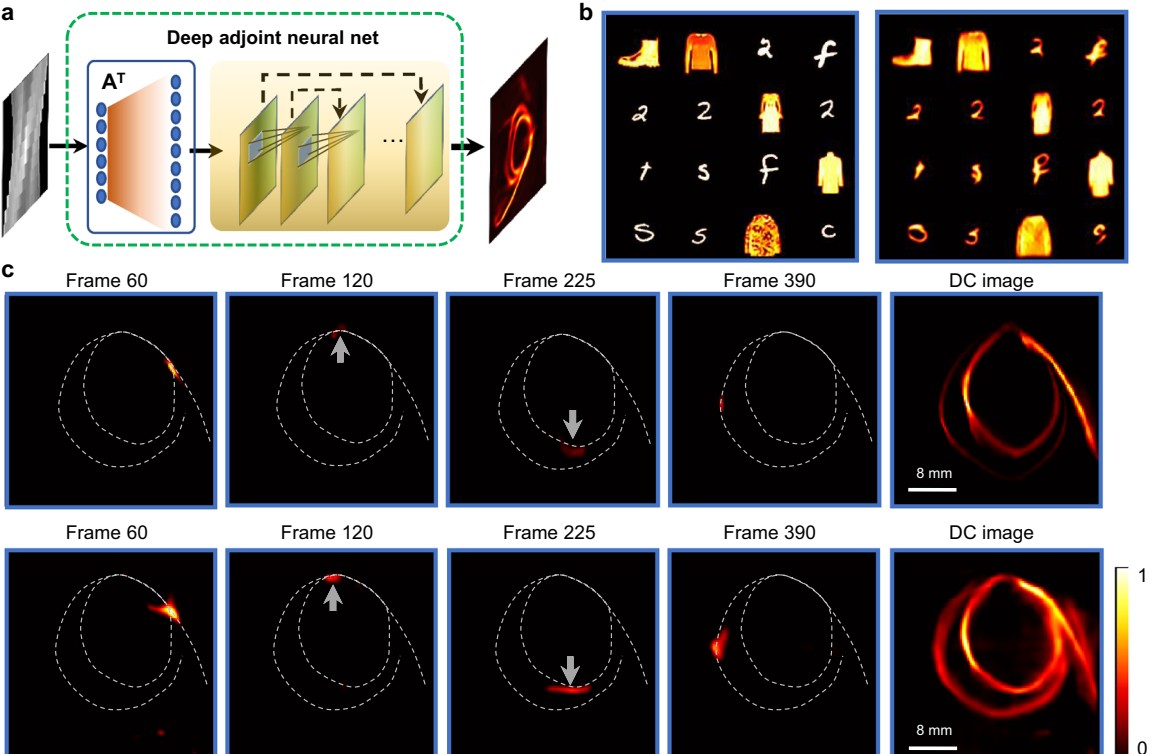

**Fig. 3 Deep adjoint neural network for LIFT reconstruction. a** The network architecture of DANN. The measurement data is firstly processed by the adjoint operator $A^T$ of the system and then passed to a convolutional network with skip connections. The detailed network structure and parameters are provided in Supplementary Note 8. **b** DANN reconstruction results of a subset of the experimentally measured test images, arranged in a montage format with an image field of view ~36 mm. Left: ground truth images; Right: DANN reconstruction. **c** Comparison of image reconstruction results using iterative methods (top row) and DANN (bottom row). The helical fiber structure is crafted and overlaid in each frame for visual guidance.

whereas the DANN manages to recover the signal decently. The lack of signals in the iterative reconstruction is more evident in the time-integrated images. Notably, the helical fiber (spanning a ~80-mm depth range) is well resolved here without the need of computational refocusing, corroborating the automatically extended depth of field.

Currently, the iterative method takes ~2.5 s to reconstruct a (128, 128, 1000) datacube when implemented on an RTX2080Ti graphical processing unit (GPU). By contrast, DANN implemented on the same GPU using PyTorch costs only ~0.5 s after training (~1.5 h), a five times speedup. The reconstruction speed can be further accelerated by efficient scaling of the neural network[38] and exploiting more powerful GPUs or alternative hardware like field programmable gate arrays for network implementation[39].

**LIFT for non-line-of-sight imaging.** Being able to acquire a large-scale 4D data cube (*x, y, u(or z), t*) with a single snapshot, LIFT stands as an enabling method for NLOS imaging at a 30 Hz video rate, which is critical for applications like navigation and surveillance. To demonstrate LIFT for NLOS imaging, we focused the camera on a diffusing wall with an FOV ~600 mm × 800 mm. A picosecond pulsed laser was collimated onto the wall, and the incident spot was blocked by a tiny stop at the relay lens' intermediate image plane to avoid the directly backscattered light from the wall. The signals from the hidden scene were recorded by LIFT with a single laser shot. With an average power at 2 mW, multiple laser shots were averaged for imaging large static scenes (total acquisition time being 0.2 s using 20 shots for objects placed ~0.3 m from the wall and 1 s using 100 shots for objects placed

>1 m from the wall). The hidden scene was then reconstructed using the extended phasor-field method (Methods section).

Figure 4a–c show the experimental imaging results of three hidden scenes: letter N, a mannequin, and two letters with occlusion. Imaging beyond the wall size is demonstrated in Fig. 4d, e for two hidden scenes, with both lateral and depth dimension stretching over 1 m. Both the shapes and 3D locations of the scenes are well reconstructed, despite that only seven projections were used in the current LIFT system. Supplementary Movie 3 illustrates the 3D reconstruction fidelity of the two letters at different depths. Supplementary Note 6 present additional results of employing LIFT for imaging complex hidden scenes on public synthetic datasets, using different number of projections under various photon budgets. The light field capabilities of LIFT can substantially lessen the focusing requirement for image acquisition, allowing non-planar walls to be exploited for NLOS imaging. We show in the last column of Fig. 4a–c the degraded hidden scene reconstruction when the camera refocused away from the wall, by undoing the automatically extended depth of field of LIFT (Supplementary Note 3.3). Although NLOS reconstruction using the phasor-field does not impose any restrictions on the geometry of the relay wall, light collection is confined to the lens system's depth of field for unambiguous separation of signals on the wall. As a result, most NLOS implementations employed a flat/slightly curved wall. Although a depth camera has the luxury of a tiny aperture (thus large depth of field) to calibrate the imaging geometry, the resultant quadratic reduction of light collection prevents similarly small apertures being used in a NLOS camera. This makes it challenging to accommodate the entire wall within NLOS camera's depth of field in real-world applications. While a recent work used dynamic

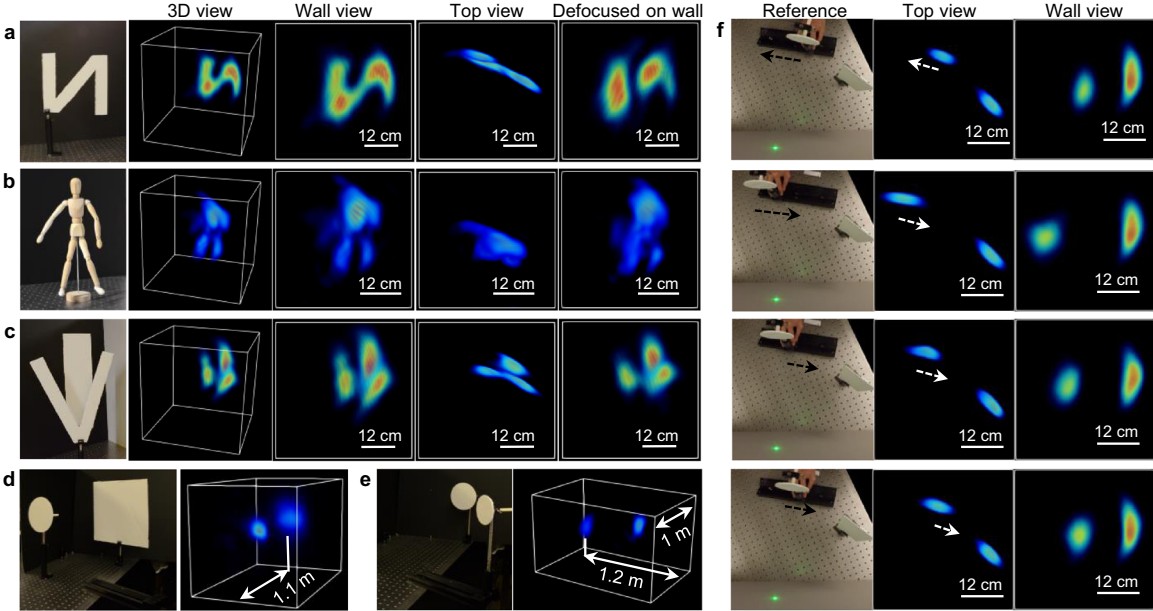

**Fig. 4 NLOS imaging by LIFT. a–c** Reconstructed images of three static hidden scenes: a letter N, a mannequin, and letters V and I at different depths, with letter I being partially occluded by V. The mannequin is reconstructed within a smaller volume than others for better visualization. The images from left to right are the reference photographs, 3D rendering of the reconstructed scenes, and images of the scene from the wall and top perspectives, respectively. The last column shows the reconstructed scene when the camera is defocused on the wall. **d, e** imaging over 1-meter scale (beyond the wall size) using 100 laser shots (total acquisition time of 1 s). The rendering dynamic range is compressed here to emphasize the object at a large distance. **f** Representative frames of NLOS imaging of a moving object at a 30 Hz video rate. The dashed arrows indicate the movement direction. The color map is 'jet' with a normalized scale.

walls[40], the light collection was still maintained at a stationary point. LIFT's automatically extended depth of field can potentially lift this restriction without any computational refocusing burden, paving the way to efficient light collection over curved or even disconnected surfaces.

To showcase video-rate NLOS imaging, we configured a hidden scene consisting of a static strip and one circular plate, which was mounted on a rail track (~0.3 m from the wall) and manually translated back-and-forth by about 150 mm within 3 s across the wall (the moving speed being ~120 mm/s or ~15% of the wall per second). LIFT recorded the 2D time-resolved data with an exposure time of 10 ns at a repetition rate of 100 Hz, and three frames were averaged to improve the SNR, yielding a frame rate of ~30 Hz. Figure 4c shows the recovered dynamic scene at different time instants along with corresponding reference photographs. LIFT captured the motion of the circular plate faithfully as compared with the reference camera. The corresponding video is provided in Supplementary Movie 4. In contrast to the previous NLOS tracking that uses a limited number of virtual sensor points[13,15,18] for localizing individual objects at a few frames per second, LIFT achieved a full 3D imaging of the hidden scene at 30 Hz, bringing NLOS imaging closer towards field deployments.

## Discussion

Although not experimentally demonstrated, we show extensive synthetic results in Supplementary Note 6 that LIFT as an imaging strategy can readily exploit a compact 1D sensor like a 1D array of SPAD detectors for high quality NLOS imaging at a 30-Hz video rate by using only a few rotations while offering unique light field capabilities. SPAD detectors feature three prominent advantages: a lower cost, a compact form factor, and a single-photon sensitivity. While 2D SPAD cameras suffer from low fill factors, 1D SPAD detectors can easily accommodate on-chip photon counters on the side of the active pixel and reach a fill

factor close to 100%[6], allowing more efficient light collection. LIFT also opens up the possibility to build 2D camera arrays with 1D sensors for ultrafast or synthetic-aperture imaging, featuring orders of magnitude smaller data load than conventional approaches. The larger baseline and etendue in camera arrays will also enable vastly larger light throughput, making it possible to see through occlusions[41].

Given its unique snapshot acquisition of a large-scale time-resolved light field data, LIFT may find a broad range of applications that are previously hindered by prolonged time-domain measurements, such as imaging into/through scattering medium via time domain diffuse optical tomography[42]. It could also be readily extended to an optical dimension other than time, such as spectral domain by using an imaging spectrometer as the 1D sensor and thereby enabling snapshot light field hyperspectral imaging. With spectral encoding being the foundation of active ultrafast cameras[1], spectral domain LIFT may turn an off-the-shelf imaging spectrometer into an ultrafast camera with sub-100-fs temporal resolution and a sequence depth over 1000, provided that an appropriate illumination is available.

## Methods

**LIFT forward model.** After resampling, LIFT is mathematically equivalent to computed tomography using parallel beam projection. Denoting the angle of the lenslet invariant axis with respect to the 1D sensor's normal as $\theta$ and the local coordinate on the sensor behind each lenslet as $k$, the 1D projection intensity $b(k, \theta)$ can be obtained by first convolving the ideal pinhole image $o(x, y)$ of the en-face object with the line-shaped PSF $\delta(x\cos\theta + y\sin\theta)$ and then sampling along the slice $y = 0$, which leads to:

$$b(k,\theta) = \left[o(x,y) * \delta(x\cos\theta + y\sin\theta)\right]_{x=k,y=0} = \iint_{-\infty}^{\infty} o(x,y)\delta[(x-k)\cos\theta + y\sin\theta]\,\mathrm{d}x\mathrm{d}y$$

$$(2)$$

where $\delta(x, y)$ is the Dirac delta function, $x$ and $y$ denote the coordinates in the image space. It is straightforward to derive from the above equation that projection along angle $\theta$ is equivalent to rotating the object by an angle of $\theta$ and then

integrating along the $y$ axis:

$$b(k,\theta) = \int\int_{-\infty}^{\infty} o(x',y')\delta(x' - k\cos\theta)\mathrm{d}x'\mathrm{d}y' = \int\int_{-\infty}^{\infty} o(x',y')\delta(x' - k')\mathrm{d}x'\mathrm{d}y'$$

(3)

where $[x',y']^T = \mathbf{R_\theta}[x,y]^T$, and $\mathbf{R_\theta}$ is the rotation matrix. $k' = k\cos\theta$ is the resampling operation as derived in Supplementary Note 1. The above equation can be discretized by sampling the object $o$ $(x, y)$ on a regular $N \times N$ grid and approximating continuous integration with finite summations. The forward model of the projection data acquisition can then be written as:

$$b(\theta) = \mathbf{T R^\theta} g$$

(4)

in which $g$ is the vectorized object image, $\mathbf{R^\theta}$ is the rotation operator, and $\mathbf{T}$ denotes the integration along the column direction of the image. The integration operator $\mathbf{T}$ can model the non-uniform intensity of the line-shaped PSF, a vignetting effect of the lenslet, which is small in photographic imaging of LIFT (Supplementary Note 1). By stacking the projection data at different angles, the forward model for LIFT with $n$ lenslets can be completed as:

$$b = \mathbf{T}\begin{bmatrix} \mathbf{R}^{\theta 1} \\ \vdots \\ \mathbf{R}^{\theta n} \end{bmatrix} g = \mathbf{A}g$$

(5)

here, $\mathbf{A}$ is the linear operator representing the system forward model.

**Image reconstruction**. Because the number of projections $n$ is generally smaller than the pixel resolution of the unknown image $N$, Eq. (5) is under-determined and hence represents a sparse view CT problem. We reconstruct the image by solving the following optimization problem:

$$\mathrm{argmin}\ \|\ b - \mathbf{A}g\ \|_2^2 + \rho\varphi(g)_1$$

(6)

where $\varphi(g)$ is a transform function sparsifying the image and $\cdot_1$ is the $l_1$ norm. $\rho$ is a hyperparameter that controls the balance between the data fidelity and regularization term. Various transform functions, like total variation, wavelet transform, and discrete cosine transform, can be used to make the image representation sparse. We chose $\varphi(g) = g$ owing to its simplicity and suitability for a massively parallel solution. Equation (6) is solved using the FISTA algorithm[43] on a GPU for optimal speeds. We found LIFT reconstruction to be relatively insensitive to the regularization parameter $\rho$: after normalizing the measurement $y$, setting $\rho$ to 0.05–0.5 leads to good results for all the experiments. For NLOS imaging, in particular, $\rho$ can span a large range (0.01–0.5) without significant influence on the reconstruction quality. This is probably attributed to the fact that the reconstruction noises and artefacts on the wall are in the 'pupil plane' of NLOS imaging.

With $n$ projections in LIFT, the complexity for reconstructing a datacube of size $(N, N, N_t)$ using $m$ iterations is $O(mnN^2N_t)$. Each iteration includes a gradient step and a simpler $l_1$ regularization step. The gradient step involves one pass of the forward operator $\mathbf{A}$ and its adjoint $\mathbf{A^T}$, both of which have a complexity of $O$ $(nN^2N_t)$: projection at each angle has a complexity of $O(N^2)$, and the $N_t$ instantaneous images are independently processed at $n$ projection angles. The regularization step has $O(N^2N_t)$ soft shrinkage operations, which is negligible in comparison. Similarly, with a depth resolution of $N_d$, the reconstruction complexity for a 3D scene $(x, y, z)$ is $O(mnN^2N_d)$: each depth is reconstructed independently after shearing the measurement data (Supplementary Note 3).

**Sparsity requirement**. The sparsity prior imposed by the regularization term in Eq. (6) may not be valid if the image to be recovered is not sufficiently compact/compressible in a chosen representation basis. Generally, the image sparsity (percentage of dominant coefficients in the basis) must be proportional to the inverse of the compression factor ($Nn^{-1}$: Nyquist sampling rate dividing the system sampling rate) in order to achieve a high-fidelity reconstruction. In Supplementary Note 5.1, we investigated LIFT for imaging scenes of different complexity under various compression factors by changing the number of projections. With a compression factor of 18 in our current implementation, LIFT can recover the low frequency structure of cluttered images but not the high frequency details. It is hence important to analyze the sparsity characteristic of the scene to be captured and choose the number of lenslets wisely to strike a balance between the image quality and resolution.

**Resolution and field of view**. The effective pixel resolution of LIFT is determined by the 1D sensor pixel number and the number of lenslet. Given $n$ lenslets and a 1D sensor with $N_x$ pixels, the effective imaging resolution for LIFT is $N = n^{-1}N_x\cos\theta_{max}$, where is $\theta_{max}$ the maximum projection angle with respect to the normal of 1D sensor, and the term $\cos\theta_{max}$ is to account for the resampling process (Supplementary Note 1). There is therefore a trade-off between the pixel resolution and image quality. The image resolution can be increased by employing fewer lenslets at the expense of reduced image quality, as the available number of projections is proportionally reduced. With seven lenslets along the streak camera's slit, the effective resolution of current LIFT camera is $128 \times 128$.

Despite the trade-off between the pixel resolution and image quality, LIFT represents a highly efficient method for light field acquisition. Using $n$ projections of $N$ pixels for reconstructing an $N \times N$ image, LIFT acquires implicitly an $n \times N \times N$ light field data ($n$ angular resolution and $N \times N$ spatial resolution) with only $n \times N$ pixels, which is $N$ times less than those of conventional (focus or unfocused) light field cameras, regardless whether the lenslet number $n$ satisfies the Nyquist sampling criterion or not. Given LIFT's spatial resolution, this fact translates to two orders of magnitude more efficient utilization of the camera pixels.

The field of view of LIFT is reduced by a factor of $n$, as the 1D sensor is divided to record the object's projection data at different angles. However, there is no inherent limit on the achievable field of view for LIFT since it is straightforward to tailor the relay systems to obtain a desired FOV for target applications.

**System photon efficiency and signal to noise ratio**. LIFT is statistically as photon-efficient as conventional cameras. The slit in LIFT is equivalent to a pixel array in conventional point-to-point imaging, not an exit/entrance aperture: a larger slit collects more light at the expense of resolution just as the pixel size did in conventional cameras. Apart from this, the light collection efficiency of LIFT is determined by its line-shaped PSF as analyzed below.

The PSF in a linear-shift-invariant system satisfies $\int\int_{-\infty}^{\infty} \mathrm{PSF}(x,y)\mathrm{d}x\mathrm{d}y = const. = 1$ That is, the light from a point source is distributed onto the sensor according to the PSF, which adds up to a constant (1 here without loss of generality). The light intensity $I$ $(x_0, y_0)$ at pixel $(x_0, y_0)$ in the image space could be written as:

$$I(x_0,y_0) = [o(x,y) * \mathrm{PSF}(x,y)]_{x=x_0,y=y_0} = \int\int_{-\infty}^{\infty} o(x,y)\mathrm{PSF}(x-x_0,y-y_0)\mathrm{d}x\mathrm{d}y$$

(7)

where $o(x, y)$ is the object and $*$ denotes the 2D convolution. Hence, each pixel records a weighted image intensity with a kernel defined by the PSF. For conventional cameras, the PSF is ideally a Dirac delta function. For LIFT, it is a uniform line spanning over the FOV. Discretizing Eq. (7) and denoting $N$ as the number of pixels inside PSF, the light collected by an individual pixel is:

$$I(x_0,y_0) = \sum_{j=1}^{N} \frac{o(x_j,y_j)}{N} = \frac{1}{N}\sum_{j=1}^{N} o(x_j,y_j)$$

(8)

where $\mathrm{PSF}(x,y) = N^{-1}$ has been used. If the object is not spatially sparse along the line-shaped PSF, the statistically expected light collection efficiency of LIFT will be the same as conventional cameras. For NLOS imaging, the instantaneous images on the wall in NLOS imaging are generally smooth but not spatially sparse, especially for complex hidden scenes as shown in Supplementary Note 6.2. As a result, the light collection efficiency of LIFT is on par with conventional camera for NLOS imaging.

Regarding the total collected light in the image space, LIFT using a 1D array of sensors records a total light of $I_{\mathrm{tot\_LIFT}} = \sum_{i=1}^{N} I(x_0,y_0) = N^{-1}\sum_{i=1}^{N}\sum_{j=1}^{N} o(x_j,y_j)$, which is $N$ times less than that collected by a 2D pixel array: $I_{\mathrm{tot\_2D}} = \sum_{i=1}^{N}\sum_{j=1}^{N} o(x_j,y_j)$. However, each pixel has its own readout and shot noises at the pixel location, elevating the total amount of noises in 2D cameras as well. On a pixel basis, the noise variance is $\sigma_{2D}^2 = o(x_j,y_j) + \sigma_G^2$ (shot noises plus readout noises) in a 2D camera and $\sigma_{\mathrm{LIFT}}^2 = N^{-1}\sum_{j=1}^{N} o(x_j,y_j) + \sigma_G^2$ in LIFT, where $\sigma_G^2$ denotes the Gaussian readout noise variance. Therefore, the statistically expected signal to noise ratio (SNR) of LIFT light collection is on par with conventional cameras:

$$E[\mathrm{SNR_{LIFT}}] = E\left[\frac{\sum_{j=1}^{N}\frac{o(x_j,y_j)}{N}}{\sigma_{\mathrm{LIFT}}}\right] = E\left[\frac{o(x_j,y_j)}{\sigma_{\mathrm{LIFT}}}\right] \approx E\left[\frac{o(x_j,y_j)}{\sigma_{2D}}\right] = E[\mathrm{SNR_{2D}}].$$

(9)

**System calibration**

*Lenslet array calibration*. To determine the angle of the invariant axis of each lenslet with respect to the slit/1D sensor, we installed a pinhole (Thorlabs P100D, 100 μm diameter) at the center of the image plane, filtering a diffused LED light to emulate a point source. The line-shaped images of the point source were subsequently captured by widely opening the slit of the streak camera without applying temporal shearing. The angles of the lenslets were found automatically using radon transformation. The center of each sub-image was directly localized from a second image of the same point source by reducing the camera slit width to 10 μm. The projection data length (sub-image size) for each lenslet was then calculated as the average pixel distance between adjacent centers. Finally, the projection data of each lenslet was extracted from the 1D sensor to form a sinogram.

*Non-ideal effects calibration*. As detailed in Supplementary Note 4, practical implementations of LIFT system suffer from non-ideal effect that can be calibrated to improve image reconstruction quality. No extra data acquisition is needed here: the point source image obtained for angle calibration suffices.

*Temporal shearing calibration.* Streak camera shows noticeable shearing distortion: the deflection onto different rows of CCD camera deviates non-uniformly across the slit from the ideal direction, which is perpendicular to the slit. We measured the actual temporal shearing by imaging a picosecond laser spot reflected from a diffusing slab and adding up the sheared images captured at different time delays. A polynomial transformation was then found to correct the shearing distortion in streak images prior to any further processing in LIFT. Still, due to the lack of a perfect modeling of the shearing distortion, small residual distortion remains that slightly degraded the temporal measurement of LIFT using streak camera.

**NLOS experiments**. The detailed system layout for NLOS experiments is illustrated in Supplementary Fig. 9. A picosecond laser (532 nm light at 100 Hz with 6 ps pulse width and 2 mW average power) was collimated onto the diffusing wall made of a white foam plate. The LIFT camera was focused at the wall with a field of view of 600 mm × 800 mm. The laser position was fixed around the center of the FOV. Ambient room light was turned on during all experiments.

*NLOS calibration.* The system's geometric configuration is measured by a structured-light depth camera. The 3D position of the wall (a dense point cloud), the laser incident spot and the LIFT camera are all obtained in the coordinate system of the depth camera. To relate each pixel of the LIFT camera to the imaged spot on the wall, a grid pattern is projected on the flat wall and imaged by both the LIFT and depth camera. The two images are registered by a Homography matrix, by which a pixel-to-pixel correspondence was established between the LIFT camera and the depth camera. Each pixel's 3D position on the wall is then identified for LIFT camera by indexing the wall's point cloud using the correspondence map.

*NLOS reconstruction.* After reconstructing the 2D time-resolved data, we unwrap the data using the calibrated geometric configuration and then reconstructed the hidden scene with the phasor-field method. To improve noise robustness of LIFT for NLOS imaging, the weighting factors[44] are extended to the phasor-field method. Under the phasor-field framework, the signals $y_r(\mathbf{r_p}, t)$ are convolved with a bandpass-filtering kernel $h(t)$ before backprojection reconstruction (the imaginary part is omitted here as it is similarly processed):

$$I(r_v, t) = \int_{-w}^{w} y_r\left(\mathbf{r_p}, t\right) * h(t - \tau) d\mathbf{r_p} \tag{10}$$

where $\mathbf{r_p}$ and $\mathbf{r_v}$ index the detection point on the wall and the reconstruction voxel respectively. $\tau = \frac{\mathbf{r_s} + \mathbf{r_p} - 2\mathbf{r_v}}{c}$ is the round-trip travel time from the illumination point $\mathbf{r_s}$ to the voxel $\mathbf{r_v}$ and back to the detection point. The coherence factor is extended here on the filtered signals:

$$CF(\mathbf{r_v}) = K^{-1} \sum_{i=1}^{K} \frac{I(\mathbf{r_v}, t = \tau + i\Delta t)}{I_q(\mathbf{r_v}, t)} \tag{11}$$

$$I_q(\mathbf{r_v}, t) = \int_{-w}^{w} \left\{ y_r\left(\mathbf{r_p}, t\right) * h(t - \tau) \right\}^2 d\mathbf{r_p} \tag{12}$$

where $K$ is the temporal kernel size, and $\Delta t$ is the time bin width. It evaluates the spatial coherence of the signals across the sampling grid: backscattered signals from the hidden objects are spatially correlated on the wall, whereas noises tend to be independent of each other. The reconstruction volume weighted by the coherence factor is then:

$$I(\mathbf{r_v}) = I(\mathbf{r_v}, t = 0)CF(\mathbf{r_v}) \tag{13}$$

The noises are attributed to the measurement shot noises, ambient light, inter-reflections and LIFT reconstruction artefacts. The ambient light is generally stable during the exposure time and can be modeled by a slowly varying function of time. Similarly, the inter-reflections tends to show as low frequency components in $y_r(\mathbf{r_p}, t)$. Therefore, a properly chosen $h(t)$ will effectively attenuate both of them. Their primary effects are on the measurement shot noises at each time bin, which are determined by the total photon count from all origins.

NLOS reconstruction using the phasor-field method has a complexity of $O(N^5)$ or $O(N^3 \log N)$ when implemented with elliptical backprojection or fast Rayleigh-Sommerfeld diffraction[19]. To accelerate computation, we implemented the phasor-field reconstruction on a GPU (Nvidia RTX2080Ti) using CUDA. For 3D rendering, the volumetric image was normalized and soft-thresholded to improve visibility. For a $128 \times 128 \times 128$ volume, the phasor-field reconstruction time is ~2.5 s. Combined with the LIFT reconstruction time of 2.5 s using iterative methods (0.5 s using DANN) at a resolution of $128 \times 128 \times 1016$, the total time of NLOS imaging is ~5.0 (or 3.0) seconds. As the computing power of GPU continues to grow, the reconstruction speed is expected to enable real-time reconstruction in the near future.

## Data availability
The experimental data of this study is available at https://github.com/Computational-Imaging-Hub/LIFT.

## Code availability
The code of this study is available at https://github.com/Computational-Imaging-Hub/LIFT.

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

## Author contributions

X.F. and L.G. conceived the study. X.F. designed and built the imaging system, performed the experiments, analyzed the data, wrote the reconstruction code, and implemented the deep adjoint neural network. L.G. contributed to the conceptual experiments design. All authors contributed to the manuscript preparation. L.G. supervised the project.

## Competing interests

The authors declare no competing interests.
