## [Peer Review File · Nature Communications]

REVIEWER COMMENTS

Reviewer #2 (Remarks to the Author):

The manuscript entitled "Ultrafast light field tomography for snapshot transient 2 and non-line-of-sight imaging" by Feng and Gao reports an interesting and novel strategy for 4D imaging. This approach converts a point in 3D space to a line using a cylindrical lens. The line-shaped point spread function allows an individual pixel to integrate the image along that line. Projection at different angles is achieved by using multiple cylindrical lenses with different orientations. To demonstrate the impact of this novel imaging strategy, the authors perform 4D imaging of a helical fiber and train a neural network to perform image refinement after the data is processed by the adjoint operator. Another interesting, perhaps more important, demonstration is for non-line-of-sight imaging at a 30 Hz video rate. A comprehensive discussion and analysis of the reported platform are provided in the supplementary materials. The presented data are solid, technically sound, and appealing. I think this manuscript is in high quality and may generate impacts for different research communities such as spectroscopy, microscopy, and computational imaging. I have the following comments that the authors may consider to better address.

- 1) It is a good demonstration using the helical fiber in Figs 2 and 3. However, it seems to me that information is very sparse in the 4D space. Essentially, the light field only occurs at one point in the 3D space at a given time point. Is it possible to demonstrate the transient light field imaging in a more complicated scene, like multiple light spots at a given time point?
- 2) The authors use 7 cylindrical lenslet array to project the images on the 1D Streak camera. However, in Fig. 1c, it seems that they use one large circular lens to relay the light to the Streak camera. I suggest they modify Fig. 1c to better reflect the use of lenslet array to avoid confusion.
- 3) The authors have discussed the use of a Dove prism to better rotate the image projection and avoid the missing cone problem. I am wondering why this scheme has not been implemented in the current setup. Adding the Dove prism before the lenslet array seems to be easy and straight forward.

Reviewer #3 (Remarks to the Author):

In general, one-dimensional sensors can achieve higher spatial and temporal resolution than 2D arrays, since acquisition and timing circuitry does not need to compete with photosensitive elements for sensor real estate. In "Ultrafast light field tomography for snapshot transient and non-line-of-sight imaging", the authors make the innovative observation that cylindrical lenses with partially vertical orientations can multiplex information onto a horizontal 1D sensor that is orthogonal to the sensor orientation, thereby recording both horizontal and vertical information. Furthermore, the spatial separation of the lenslets along the 1D sensor baseline provides multiview information, thus encoding a light field. Finally, using a 1D time-resolved sensor, the authors are able to reconstruct 4D information: 3 spatial dimensions plus time. Because the particular 1D sensor is a streak camera, the technique is remarkably sensitive enough that a single snapshot is capable of recovering the complete 4D data.

The paper describes a novel acquisition approach that could potentially have a significant impact in numerous fields that require fast 3D imaging. For this reason, the paper would be a good fit for Nature Communications. However, there are a number of points that should be addressed or more clearly explained first.

Primarily, it would be helpful to be more clear about what acquisition components and analysis are used for 2D, 3D, and 4D information. For example, multiple 1D projections yields a sparse view CT problem (2D). Capturing these 1D projections from different perspectives (multi-view stereo) enables light-field (3D) capture. The time dimension is achieved by making the same type of measurement over time. In particular, the multi-perspective effect of the cylindrical lenslets at

different positions leading to light field information is not well explained.

Secondly, it is important to more carefully address the requirements on the sparsity of the scene, and whether the approach could fare well for a more "cluttered" scene.

A number of detailed questions and suggested references are listed below.

- Lines 19-20: There seems to be something missing here---is it a temporal sequence of over 1000 frames? The same unclear wording is used in line 42.

- Lines 36 - 39:

-- References 5 and 7 describe devices (ICCD and SPAD array) that have no inherent time resolution, so temporally scanning a gate window is necessary to capture a time-resolved image. However, many SPAD arrays are designed to perform time-correlated single photon counting (TCSPC), which time stamps individual photon detections and does not require temporal scanning of a gate.

-- Although fitting time-to-digital convertors and photosensitive elements on the same sensor chip can be challenging, new detector architectures (e.g., those using 3D stacking) are improving fill factor and enabling larger-format arrays with time resolution for each pixel. Some recent works are listed below.

-- It is correct that the scope of these camera is limited to repeatable events (not single-shot), but instead of "temporal scanning", this is because most TCSPC designs can detect only one photon at a time and thus require repeated illumination to build up a sufficient histogram of photon detection times to capture transient behavior.

[1] A. MacCarone, F. Mattioli Della Rocca, A. McCarthy, R. Henderson, and G. S. Buller, "Three-dimensional imaging of stationary and moving targets in turbid underwater environments using a single-photon detector array," *Opt. Express*, vol. 27, no. 20, p. 28437, Sep. 2019.

[2] R. K. Henderson et al., "A 192 x 128 Time Correlated SPAD Image Sensor in 40-nm CMOS Technology," *IEEE J. Solid-State Circuits*, vol. 54, no. 7, pp. 1907-1916, Jul. 2019.

[3] C. Zhang, S. Lindner, I. M. Antolovic, J. Mata Pavia, M. Wolf, and E. Charbon, "A 30-frames/s, 252 x 144 SPAD Flash LiDAR with 1728 Dual-Clock 48.8-ps TDCs, and Pixel-Wise Integrated Histogramming," *IEEE J. Solid-State Circuits*, vol. 54, no. 4, pp. 1137-1151, Apr. 2019.

[4] I. Gyongy et al., "High-speed 3D sensing via hybrid-mode imaging and guided upsampling," *Optica*, vol. 7, no. 10, p. 1253, Aug. 2020.

- Lines 59 - 64:

--The acquisition times of NLOS methods depend on a number of factors. Many approaches take a long time simply because the detected signal is very weak, so longer acquisition times allow for the accumulation of more photons. In addition, confocal methods require independently scanning a sequence of co-located illumination and detection points, so the acquisition time depends on the number of scanned points (i.e., the reconstruction resolution).

One approach to speeding up NLOS imaging has been to employ higher-powered lasers and occasionally also retro-reflective scene elements, so that more photons can be detected in a shorter period of time.

--Using a high-powered laser, Ref. 17 (Lindell, Wetzstein, & O'Toole, "Wave-based Non-line-of-sight Imaging Using Fast F-k Migration") was still able to achieve 4 Hz 32x32-pixel videos of a person in a retroreflective suit.

--Another preprint (J. H. Nam, E. Brandt, S. Bauer, X. Liu, E. Sifakis, and A. Velten, "Real-time Non-line-of-Sight imaging of dynamic scenes." arXiv: 2010.12737, 2020.) recently demonstrated 5 frame-per-second videos for large-scale non-retroreflective scenes, but also with a high powered laser. Note that there are some superficial similarities with this submission in using a 1D sensor and an extension of the phasor field method. It would be useful to have some commentary on how the methods differ.

-Lines 73-83

This paragraph introducing the topic seems like it could fit better as the last paragraph of the previous section (the Introduction) as a general overview of the solution.

Also, 75-77 is somewhat confusing: "exploiting the fact that 1D sensors are vastly faster" implies that the temporal dimension is necessary for converting "a 1D sensor to a 2D light field camera." But that process of converting a 1D measurement into a 2D light field does not require temporal resolution---only spatially multiplexing multiple tomographic acquisitions onto different parts of the sensor and using the spatial disparity between the different acquisitions for depth. It would be helpful to be more clear about which components enable different parts of the acquisition (e.g., what is required to get 2D, 3D, and 4D (3D + time))?

-Line 83: Is there anything inherent about LIFT acquisition that enables NLOS with a lower laser power? As discussed before, the higher laser power is typically for larger scale scenes and/or to acquire more light from fewer repetitions. How could LIFT scale to larger scenes, e.g., 3m x 3m x 3m?

-Line 91: Part b of the figure and caption for Figure 1 could be more informative. The term "PSF substitution" is particularly unclear. Instead, it could be better described as convolving the object with the linear PSF of the lenslet, then sampling along the horizontal axis of the 1D sensor. Finally, it would be useful to describe how the temporal domain is captured.

-Lines 138-139:

The sentence "The synergy between compressive data acquisition and deep neural network breaks the data bandwidth limit of conventional cameras..." seems to overstate the role of the neural network. The NN can apply strong shape priors for compressive sensing reconstruction, but it is only the sparsity of the scene in a relevant basis that allows that reconstruction to succeed. Hence the success of the iterative methods as well.

- Lines 153 - 158

The experiment of imaging the light pulse as it propagates through the twisted fiber is reminiscent of two papers using linear SPAD arrays. It would be useful to comment on differences in the resolution (spatial and temporal), acquisition time, and requirement of a sparse scene (not necessary for the direct-detection SPAD arrays, but perhaps critical for the compressive light-field acquisition).

[1] M. O'Toole et al., "Reconstructing Transient Images from Single-Photon Sensors," in Proc. of CVPR, 2017, pp. 2289-2297.

[2] D. B. Lindell, M. O'Toole, and G. Wetzstein, "Towards transient imaging at interactive rates with single-photon detectors," in International Conference on Computational Photography, 2018.

-Fig. 2: The transverse dimension of the fiber acquisition is extremely small. Is that a limitation of the method?

Also, part (d) is hard to follow---could the images be generated with a rescaled or different colormap so that the pulse is more clearly visible?

-Fig. 3: Is the training data simulated to appear at different depths or all at the same plane?

Also, the iterative method actually seems a bit less blurry, although dimmer, than the DANN reconstructions.

Methods

-Lift Forward Model: Equation (2) combines a number of mathematical abstractions simultaneously. It could be helpful to back up slightly and first present the model as 1) convolving the object with a linear point spread function and 2) sampling the convolved result along the horizontal axis. This can be expressed before writing out the double integral that performs this operation.

-Line 358: Can you comment on the necessity of the scenes to be sparse? This is clearly encoded in the choice of $\phi(g) = g$, but is not directly addressed in the main paper. However, it is clear that the light in the fiber, the NLOS scenes, and the training data for the NN inverse are all highly sparse. This helps explain why sparsity in various transform domains would not make a significant difference. How complicated can the scene get though? Would TV regularization be enough for more complex scenes that do not simply have a bright patch of interest surrounded by a black background?

-Lines 366-372: This analysis appears to be only for reconstructing 2D images over time. Since each time slice is processed independently, the analysis could be shown just for a single 2D reconstruction. Is there also analysis for the light field reconstruction (from different perspectives)?

Supplementary Information

-Eqn. S5 is key to answering some of my questions about how the light field reconstruction can be performed (due to the u term) and may be helpful to include in the main text as well.

-Fig. S2- is there a panel (g) missing?

-Lines 91-92: What does it mean "Actual implementation of LIFT usually restricts the number of projections in lieu of ten"? Is it supposed to imply the number of projections is restricted to less than ten?

-Line 140: "LIFT captures only the angular information along one axis (Q) instead of two"---what effect does this have on 3D reconstruction? The results show that depth can be reconstructed with only 1 dimension of angular information. How much would the second axis help?

-S3.3: It would be great to include the actual optimized lenslet angles for the depth-of-field and depth-sense versions of the lenslet arrangements for the 7-lenslet implementations.

-Line 213: What metric is used as a focus measure to be able to infer depth?

- Is there any section that details the actual experimental hardware used (e.g., manufacturer, basic specifications, etc.)?

Response Letter

The authors thank the reviewers for their insightful comments, which have greatly improved the quality of our manuscript. In the following, we provide point-by-point responses. The changes in the text are highlighted in red.

Reviewer 1

The manuscript entitled “Ultrafast light field tomography for snapshot transient 2 and non-line-of-sight imaging” by Feng and Gao reports an interesting and novel strategy for 4D imaging. This approach converts a point in 3D space to a line using a cylindrical lens. The line-shaped point spread function allows an individual pixel to integrate the image along that line. Projection at different angles is achieved by using multiple cylindrical lenses with different orientations. To demonstrate the impact of this novel imaging strategy, the authors perform 4D imaging of a helical fiber and train a neural network to perform image refinement after the data is processed by the adjoint operator. Another interesting, perhaps more important, demonstration is for non-line-of-sight imaging at a 30 Hz video rate. A comprehensive discussion and analysis of the reported platform are provided in the supplementary materials. The presented data are solid, technically sound, and appealing. I think this manuscript is in high quality and may generate impacts for different research communities such as spectroscopy, microscopy, and computational imaging. I have the following comments that the authors may consider to better address.

Response:

We thank the reviewer for the positive and constructive comments on this work. We addressed the comments in detail below.

1. It is a good demonstration using the helical fiber in Figs 2 and 3. However, it seems to me that information is very sparse in the 4D space. Essentially, the light field only occurs at one point in the 3D space at a given time point. Is it possible to demonstrate the transient light field imaging in a more complicated scene, like multiple light spots at a given time point?

Response:

Yes, it was demonstrated in Figure S5d of **Supplementary Materials**: because the space is decoupled from the time dimension in LIFT, 3D (x, y, z) reconstruction at each time point is independently performed. Figure S5d shows the 3D imaging results of a 3×3 grid pattern on a slanted plane in the static mode (transient scene at different time points are the same). The results are appended below for reference.

Figure S5 d and e, 3D rendering of a slanted plane displaying a grid pattern of dots.

2. The authors use 7 cylindrical lenslet array to project the images on the 1D Streak camera. However, in Fig. 1c, it seems that they use one large circular lens to relay the light to the Streak camera. I suggest they modify Fig. 1c to better reflect the use of lenslet array to avoid confusion.

Response:

We revised Fig. 1c to emphasize the utilization of a cylindrical lenslet array by changing the 'Camera lens' to 'Relay lens' and zoomed into the lenslet array. The figure caption is also revised to indicate this point. We appended the revised Figure 1 and its caption below.

Fig. 1. Working principle and implementation of light field tomography. a, Illustration of image formation by a cylindrical lens. Three point sources in the object space are transformed into parallel lines on the image plane, producing a projection image. Acquiring such projection images from different perspectives using lenslets oriented at different angles naturally samples the light field of the 3D scene, as exemplified in the insets P1-P3, where the image centre is highlighted to visualize the disparities. b, Two-step modelling of cylindrical lenslet imaging process. For clarity, an image showing predominantly point-like structures is rendered. The 1D projection data is obtained by sampling the convolution result of the pinhole image and line-shaped PSF. Recording such 1D data over time yields a time-resolved measurement. c, Typical system setup of a LIFT camera. The cylindrical lenslet array is closely secured to the entrance slit of the streak camera.

3. The authors have discussed the use of a Dove prism to better rotate the image projection and avoid the missing cone problem. I am wondering why

this scheme has not be implemented in the current setup. Adding the Dove prism before the lenslet array seems to be easy and straight forward.

Response:

We thank the reviewer for this point. The major downside of the Dove prism is that it will induce astigmatism for non-collimated light. Also, it will introduce chromatic aberrations for broadband scenes. Considering the pros and cons, we think it is justified to leave this idea to future exploration.

Regarding this, we revised in section 2.2 of Supplementary Materials to clarify tis point that reads “... The downside of using a Dove prism is that it introduces astigmatism for non-collimated light and chromatic aberrations for broadband scenes, compromising the 3D imaging performance of LIFT.”

Reviewer 2

In general, one-dimensional sensors can achieve higher spatial and temporal resolution than 2D arrays, since acquisition and timing circuitry does not need to compete with photosensitive elements for sensor real estate. In “Ultrafast light field tomography for snapshot transient and non-line-of-sight imaging”, the authors make the innovative observation that cylindrical lenses with partially vertical orientations can multiplex information onto a horizontal 1D sensor that is orthogonal to the sensor orientation, thereby recording both horizontal and vertical information. Furthermore, the spatial separation of the lenslets along the 1D sensor baseline provides multiview information, thus encoding a light field. Finally, using a 1D time-resolved sensor, the authors are able to reconstruct 4D information: 3 spatial dimensions plus time. Because the particular 1D sensor is a streak camera, the technique is remarkably sensitive enough that a single snapshot is capable of recovering the complete 4D data.

The paper describes a novel acquisition approach that could potentially have a significant impact in numerous fields that require fast 3D imaging. For this reason, the paper would be a good fit for Nature Communications. However, there are a number of points that should be addressed or more clearly explained first.

Response:

We appreciate the reviewer’s extensive and constructive comments on our work. Extensive revisions have been made on the manuscript and supplementary materials accordingly to address the raised points, as detailed below.

1. Primarily, it would be helpful to be more clear about what acquisition components and analysis are used for 2D, 3D, and 4D information. For example, multiple 1D projections yields a sparse view CT problem (2D). Capturing these 1D projections from different perspectives (multi-view stereo) enables light-field (3D) capture. The time dimension is achieved by making the

same type of measurement over time. In particular, the multi-perspective effect of the cylindrical lenslets at different positions leading to light field information is not well explained.

Response:

To address the reviewer’s concern, we detailed the process of 2D, 3D, and 4D reconstruction in Section 3.4 of the revised **Supplementary Materials** and **Supplementary Figure S3c**, which is excerpted below for reference.

“The processing pipeline of LIFT for reconstructing multi-dimensional images (2D, 3D, and 4D) is summarized in Fig. S3c. Each 1D measurement data is ordered into a sinogram (the projection data (x, θ)), which can be directly reconstructed into a 2D (x, y) image or go through a shear-and-reconstruct process to refocus on different depths, producing a focal stack. Afterwards, the focal stack is co-registered because the refocusing can induce image shifts, as explained in the previous section. The denoising algorithm VBM3D is then applied to attenuate the refocusing artefacts in the focal stack, which substantially improves the robustness of depth retrieval. Finally, the focus measure is computed for each pixel, and a quick sorting algorithm identifies the correct focal setting and maps that pixel to the corresponding depth, yielding the 3D image (x, y, z) . Owing to the decoupled space-time acquisition in LIFT, the 2D and 3D images processing are independently performed at each time instance to produce the final 3D (x, y, z) or 4D (x, y, z, t) results.”

Figure S3c, Processing pipeline for 2D (x, y) and 3D (x, y, z) imaging in LIFT. For 4D (x, y, z, t) imaging, the 3D image processing is individually applied at each time instance.

Also, as suggested by the reviewer in comments 18 (first comment for **Supplementary Materials**), we incorporated the light field data acquisition Equation S5 into the main text to explain the process of light field acquisition that reads “ Formally, as analyzed thoroughly in Section 1 of **Supplemental Materials**, the light field data acquisition of LIFT can be encapsulated into a single equation: ignoring image magnification, the projected coordinate of a point source located at (x_0, y_0) is $x_l = -x_0 - y_0 \tan \theta + u$ on the 1D sensor, where u denotes the angular component contributed by the lenslet array and θ is the orientation angle of the lenslet. The acquired projection data in LIFT relates to the *en-face* object via the Fourier slice theorem after computational resampling.”

2. Secondly, it is important to more carefully address the requirements on the sparsity of the scene, and whether the approach could fare well for a more “cluttered” scene.

Response:

We appreciate the reviewer for this advice. While LIFT can in theory satisfy the Nyquist sampling criterion, it is often not the case due to the limited pixel count of 1D sensors (Section 2.2 of **Supplementary Materials**). As a result, our current LIFT implementation indeed requires a sparsity prior in a representation basis for image recovery. We added a dedicated subsection in **Methods** to discuss this point, as appended below.

“Sparsity requirement. The sparsity prior imposed by the regularization term in Eq. (6) may not be valid if the image to be recovered is not sufficiently compact/compressible in a chosen representation basis. Generally, the image sparsity (percentage of dominant coefficients in the basis) must be proportional to the inverse of the compression factor (N/n : Nyquist sampling rate dividing the system sampling rate) in order to achieve a high-fidelity reconstruction. In Section 5.1 of **Supplementary Materials**, we investigated LIFT for imaging scenes of different complexity under various compression factors by changing the number of projections. With a compression factor of 18 in our current implementation, LIFT can recover the low frequency structure of cluttered images but not the high frequency details. It is hence important to analyze the sparsity characteristic of the scene to be captured and choose the number of lenslets wisely to strike a balance between the image quality and resolution.”

Section 5 of **Supplementary Materials** discusses the factors that affect the imaging quality of LIFT and we added in **Figure S7** a cluttered camera-man photograph to demonstrate the reconstruction quality of LIFT. **The revised Section 5.1** is excerpted below for reference.

“Section 5.1 **Compression factor.** ... Sampled at the Nyquist rate, the images recovered with a projection number of 128 serves as the ground truth **reference for calculating the peak signal to noise ratio (PSNR) of other reconstructed images.** It is noted that, as the compression factor gets larger (i.e., fewer projections), **the PSNR of the reconstructed images becomes smaller and fine image details gradually get washed out.** Moreover, the cluttered camera-man photograph renders a smaller PSNR than that of the Shepp-Logan phantom when employing the same compression factor. Therefore, the number of projections must be appropriately scaled to accommodate scenes of different complexity.”

Figure S7. LIFT image reconstruction using different number of projections for **a**, Shepp-Logan phantom and **b**, the cluttered camera-man photograph, both at a resolution of 128x128. The compression factor varies from ~18 to 1 (Nyquist rate) when the projection number changes from 7 to 128. **c** and **d**, The PSNR of the reconstructed images versus the compression factor (1 not included as it corresponds to the reference image) for the phantom and camera-man photograph, respectively.

Specific comments

A number of detailed questions and suggested references are listed below.
 3. Lines 19-20: There seems to be something missing here---is it a temporal sequence of over 1000 frames? The same unclear wording is used in line 42.

Response:

We corrected these two omissions in the revised manuscript to read “a temporal sequence of over 1000”.

4. Lines 36 - 39: References 5 and 7 describe devices (ICCD and SPAD array) that have no inherent time resolution, so temporally scanning a gate window is necessary to capture a time-resolved image. However, many SPAD arrays are designed to perform time-correlated single photon counting (TCSPC), which time stamps individual photon detections and does not require temporal scanning of a gate.

Response:

We agree that many SPADs don't scan a gate but rather rely on repeated illuminations to build up a histogram, which is another form of 'temporal scanning'. To clarify it, we revised the sentence to “temporal scanning or repeated illuminations” in the manuscript.

5. Although fitting time-to-digital convertors and photosensitive elements on the same sensor chip can be challenging, new detector architectures (e.g., those using 3D stacking) are improving fill factor and enabling larger-format arrays with time resolution for each pixel. Some recent works are listed below.

-- It is correct that the scope of these camera is limited to repeatable events (not single-shot), but instead of “temporal scanning”, this is because most TCSPC designs can detect only one photon at a time and thus require repeated illumination to build up a sufficient histogram of photon detection times to capture transient behavior.

[1] A. Maccarone, F. Mattioli Della Rocca, A. McCarthy, R. Henderson, and G. S. Buller, “Three-dimensional imaging of stationary and moving targets in turbid underwater environments using a single-photon detector array,” *Opt. Express*, vol. 27, no. 20, p. 28437, Sep. 2019.

[2] R. K. Henderson et al., “A 192 x 128 Time Correlated SPAD Image Sensor in 40-nm CMOS Technology,” *IEEE J. Solid-State Circuits*, vol. 54, no. 7, pp. 1907–1916, Jul. 2019.

[3] C. Zhang, S. Lindner, I. M. Antolovic, J. Mata Pavia, M. Wolf, and E. Charbon, “A 30-frames/s, 252 x 144 SPAD Flash LiDAR with 1728 Dual-Clock 48.8-ps TDCs, and Pixel-Wise Integrated Histogramming,” *IEEE J. Solid-State Circuits*, vol. 54, no. 4, pp. 1137–1151, Apr. 2019.

[4] I. Gyongy et al., “High-speed 3D sensing via hybrid-mode imaging and guided upsampling,” *Optica*, vol. 7, no. 10, p. 1253, Aug. 2020.

Response:

We thank the reviewer for pointing out these references on 2D SPAD sensors and agree that 3D stacking can improve their fill factors. Nevertheless, the highest fill factors of the 2D SPAD array in [4] is about 50% and the counters were shared among 4x4 SPADs (64x64 counters for 256x256 pixels), which is still inferior to that of 1D SPAD sensors. We cited reference [4] and tone-down the sentence to read “However, obtaining a grayscale time-resolved data still requires temporal scanning or repeated illuminations with a time correlated single photon counter (TCSPC), which leads to an inferior filling factor for 2D SPAD sensors⁸ given current fabricating technologies.”

6. Lines 59 - 64: --The acquisition times of NLOS methods depend on a number of factors. Many approaches take a long time simply because the detected signal is very weak, so longer acquisition times allow for the accumulation of more photons. In addition, confocal methods require independently scanning a sequence of co-located illumination and detection points, so the acquisition time depends on the number of scanned points (i.e., the reconstruction resolution).

One approach to speeding up NLOS imaging has been to employ higher-powered lasers and occasionally also retro-reflective scene elements, so that more photons can be detected in a shorter period of time.

--Using a high-powered laser, Ref. 17 (Lindell, Wetzstein, & O'Toole, "Wave-based Non-line-of-sight Imaging Using Fast F-k Migration") was still able to achieve 4 Hz 32x32-pixel videos of a person in a retroreflective suit.

--Another preprint (J. H. Nam, E. Brandt, S. Bauer, X. Liu, E. Sifakis, and A. Velten, "Real-time Non-line-of-Sight imaging of dynamic scenes." arXiv: 2010.12737, 2020.) recently demonstrated 5 frame-per-second videos for large-scale non-retroreflective scenes, but also with a high powered laser. Note that there are some superficial similarities with this submission in using a 1D sensor and an extension of the phasor field method. It would be useful to have some commentary on how the methods differ.

Response:

Compared to the 2D point scanning in Lindell's work and one-axis spatial scanning using a 1D sensor array (parallelized acquisition) in Nam's recent work, which also employed a sparse acquisition (illumination) strategy for speeding up, LIFT acquisition is faster because it eliminates the need of spatial scanning. We comment on these methods adopted in the two works in line 67-71 of the revised manuscript that reads " ...Faster scanning can also be achieved in several other ways: shortening the sensor exposure time, reducing the spatial scanning density, or parallelizing acquisition²³. Nevertheless, the scanning mechanism still persists, and the resultant smaller photon counts from shorter exposure typically need to be compensated by using a higher laser power and/or retro-reflective targets¹⁸."

The reconstruction method employed in Nam's work was the published fast frequency domain phasor field method, which was cited in our manuscript. The extension was for remapping their NLOS data in order to make the frequency domain method applicable, not an extension on the phasor field algorithm itself. Our extension on the time-domain phasor field method was detailed in Methods.

7. Lines 73-83 This paragraph introducing the topic seems like it could fit better as the last paragraph of the previous section (the Introduction) as a general overview of the solution.

Response:

We moved the paragraph to the introduction part.

8. Also, 75-77 is somewhat confusing: "exploiting the fact that 1D sensors are vastly faster" implies that the temporal dimension is necessary for converting "a 1D sensor to a 2D light field camera." But that process of converting a 1D measurement into a 2D light field does not require temporal resolution---only spatially multiplexing multiple tomographic acquisitions onto different parts of the sensor and using the spatial disparity between the different acquisitions for depth. It would be helpful to be more clear about which components enable different parts of the acquisition (e.g., what is required to get 2D, 3D, and 4D (3D + time))?

Response:

We addressed this comment in our response to the reviewer's comment 1. Also, we clarify the sentence in the revised Manuscript to read "...This is achieved by transforming a one-dimensional (1D) sensor to a 2D light field camera, exploiting the fact conventional light field acquisition is highly redundant—the sub-aperture images are mostly the same except for disparity cues. **The vastly faster frame rate of 1D sensors also benefits LIFT for high speed imaging...**".

9. Line 83: Is there anything inherent about LIFT acquisition that enables NLOS with a lower laser power? As discussed before, the higher laser power is typically for larger scale scenes and/or to acquire more light from fewer repetitions. How could LIFT scale to larger scenes, e.g., 3m x 3m x 3m?

Response:

A higher laser power is indeed the most effective solution to image larger scenes such as 3 m × 3 m × 3 m. LIFT can accommodate lower laser power due to its single-shot acquisition of the (x, y, t) datacube: given the same total acquisition time, this translates to much longer exposure time than scanning-based solutions using a single SPAD or 1D SPAD array. This is discussed in section 6.3 of Supplementary Materials. A second reason, not related to LIFT acquisition method, is that the streak camera can acquire the complete temporal trace with a single laser shot while SPAD's dead time can waste a lot of photons when the photo counts are significantly more than 1. To scale to 3 m from current LIFT-NLOS experiments, a higher laser power ($3^4 \times 2 \text{ mW} = 160 \text{ mW}$) will be needed.

Regarding this, we clarified in the Section 7 of the revised Supplementary Materials that reads " ... **The snapshot acquisition enables** LIFT to achieve drastically faster NLOS imaging with a resolution and quality close to those in dense point-scanning methods, allowing a low laser power to be used for imaging over 1 m scale. **By scaling according to the r^4 photon decay law in NLOS imaging, LIFT is expected to reach an imaging volume around 3 m × 3 m × 3 m with an average laser power of 160 mW.**"

10. Line 91: Part b of the figure and caption for Figure 1 could be more informative. The term "PSF substitution" is particularly unclear. Instead, it could be better described as convolving the object with the linear PSF of the lenslet, then sampling along the horizontal axis of the 1D sensor. Finally, it would be useful to describe how the temporal domain is captured.

Response:

We appreciate this advice. We changed 'PSF substitution' to 'PSF convolution' to better match with the mathematical modelling of LIFT image formation throughout the manuscript (and Supplementary Materials).

Regarding this, we explained both the modelling of LIFT image formation and the acquisition of temporal information in the revised Figure 1 and its caption, as appended below.

Fig. 1. Working principle and implementation of light field tomography. **a**, Illustration of image formation by a cylindrical lens. Three point sources in the object space are transformed into parallel lines on the image plane, producing a projection image. Acquiring such projection images from different perspectives using lenslets oriented at different angles naturally samples the light field of the 3D scene, as exemplified in the insets P1-P3, where the image centre is highlighted to visualize the disparities. **b**, Two-step modelling of cylindrical lenslet imaging process. For clarity, an image showing predominantly point-like structures is rendered. **The 1D projection data is obtained by sampling the convolution result of the pinhole image and line-shaped PSF. Recording such 1D data over time yields a time-resolved measurement.** **c**, Typical system setup of a LIFT camera. **The cylindrical lenslet array is closely secured to the entrance slit of streak camera.**

11. Lines 138-139: The sentence “The synergy between compressive data acquisition and deep neural network breaks the data bandwidth limit of conventional cameras...” seems to overstate the role of the neural network. The NN can apply strong shape priors for compressive sensing reconstruction, but it is only the sparsity of the scene in a relevant basis that allows that reconstruction to succeed. Hence the success of the iterative methods as well.

Response:

Compressive sensing typically required time-consuming reconstruction that slows down the imaging speed. We clarify this point to indicate that it is the accelerated reconstruction speed by DANN that improves the actual data-bandwidth of cameras, which reads “... The synergy between compressive data acquisition and **fast** deep neural network **reconstruction** breaks the data bandwidth limit of conventional cameras and enables high-resolution 2D imaging with 1D sensors.”

12. Lines 153 – 158 The experiment of imaging the light pulse as it propagates through the twisted fiber is reminiscent of two papers using linear SPAD arrays. It would be useful to comment on differences in the resolution (spatial and temporal), acquisition time, and requirement of a sparse scene (not

necessary for the direct-detection SPAD arrays, but perhaps critical for the compressive light-field acquisition).

[1] M. O’Toole et al., “Reconstructing Transient Images from Single-Photon Sensors,” in Proc. of CVPR, 2017, pp. 2289–2297.

[2] D. B. Lindell, M. O’Toole, and G. Wetzstein, “Towards transient imaging at interactive rates with single-photon detectors,” in International Conference on Computational Photography, 2018.

Response:

The demonstration of light pulse propagation of LIFT differs from the two works in two major aspects— single-shot acquisition and light field capabilities, which endow LIFT with an extended depth-of-field and 4D imaging (3D space and time). Both works relied on mechanical scanning of a 1D SPAD sensors.

Regarding this, we cited the two related works in Line 197 of the revised manuscript “...Such extended depth of field and 3D imaging capabilities are defining features of LIFT over other 2D ultrafast cameras^{33,34} (Section 7 of Supplementary Materials).”

We compare LIFT with them more comprehensively in Section 7 of the revised Supplementary Materials as below.

“While SPAD cameras^{16,17} can acquire high-resolution images at the Nyquist rate and, therefore, accommodate cluttered natural scenes better, the need of spatial scanning and repeated illuminations leads to prolonged acquisition. Interestingly, the transient images at each time instant obtained by SPAD cameras^{16,17} also show notable compressibility—they are far simpler than the static photograph of the cluttered scene, which will be accentuated with a higher temporal resolution.”

Table S1 Comparison of transient imaging performance by various methods

Methods	Resolution	Temporal resolution	Sequence depth	Compression factor	Light field	Active illumination	Scanning
STEAM ¹²	50×50	> 10 ns	Continuous	NA	No	Yes	No
STAMP ¹³	450×450	~ 230 fs	6	NA	No	Yes	No
FRAME ¹⁴	512×512	~ 125 fs	4	NA	No	Yes	No
CUP ¹⁵	150×150	~ 10 ps	350	~ 100	No	No	No
LIFT	128×128	< 10 ps	> 1000	~ 18	Yes	No	No
SPAD ¹⁶	320×240	~ 300 ps	~ 300	NA	No	No	Yes (64 s)
SPAD ¹⁷	256×250	~ 300 ps	> 1000	NA	No	No	Yes (1 s)

13. Fig. 2: The transverse dimension of the fiber acquisition is extremely small. Is that a limitation of the method? Also, part (d) is hard to follow---could the images be generated with a rescaled or different colormap so that the pulse is more clearly visible?

Response:

It is not a limitation of the method since the field of view (FOV) can be tailored by changing the image magnification either by moving the lenslet array closer/further to the camera slit or adding a relay system. For instance, the FOV of the LIFT camera is made to 600 mm × 800 mm on the wall in NLOS imaging experiments. We opted a small FOV for imaging the fiber because a large scene with an extended depth-of-

field will need a long light-diffusing fiber, which will make the single-shot data acquisition very noisy due to the exponential light decay inside the light-diffusing fiber.

We revised Fig. 2d to make the pulse more visible as below.

Fig. 2. Transient light field imaging by a LIFT camera. d, 4D (3D space and time) characterization of a picosecond laser pulse propagating inside a light diffusing fiber. The helical fiber structure is crafted and overlaid in each 3D frame for visual guidance.

14. Fig. 3: Is the training data simulated to appear at different depths or all at the same plane? Also, the iterative method actually seems a bit less blurry, although dimmer, than the DANN reconstructions.

Response:

The experimentally acquired training dataset is displayed on a monitor, therefore all at the same plane (nominal image plane). Training the network at different depths may be a way to extend the depth-of-field as an otherwise defocused image might be recovered to a focused one by a network trained this way, which we didn't explore in this manuscript. Regarding this, caption of Figure S14 stated that "... For training data acquisition, the LIFT camera captures (without temporal deflection) the training images streamed on a high-resolution monitor in a synchronized manner..."

The DANN reconstruction maybe a bit blurry as there is an imperfect misalignment correction during experimental DANN training: the centre of ground-truth images on the monitor does not coincide with the image centre of the LIFT camera. Before forming the measurement and ground-truth image pairs for DANN training, we extracted and corrected this misalignment by imaging and co-registering a grid pattern with its ground truth.

Methods

15. Lift Forward Model: Equation (2) combines a number of mathematical abstractions simultaneously. It could be helpful to back up slightly and first present the model as 1) convolving the object with a linear point spread function and 2) sampling the convolved result along the horizontal axis. This can be expressed before writing out the double integral that performs this operation.

Response:

We add more details on Equation 2 in Methods to elaborate the modelling process as excerpted below.

“ Denoting the angle of the lenslet invariant axis with respect to the 1D sensor’s normal as θ and the local coordinate on the sensor behind each lenslet as k , the 1D projection intensity $b(k, \theta)$ can be obtained by first convolving the ideal pinhole image $o(x, y)$ of the *en-face* object with the line-shaped PSF $\delta(x\cos\theta + y\sin\theta)$ and then sampling along the slice $y = 0$, which leads to:

$$b(k, \theta) = [o(x, y) * \delta(x\cos\theta + y\sin\theta)]_{x=k, y=0} = \iint_{-\infty}^{\infty} o(x, y) \delta[(x - k)\cos\theta + y\sin\theta] dx dy \quad (2)$$

where $\delta(x, y)$ is the Dirac delta function, x and y denote the coordinates in the image space. It is straightforward to derive from the above equation that projection along angle θ is equivalent to rotating the object by an angle of θ and then integrating along the y axis:

$$b(k, \theta) = \iint_{-\infty}^{\infty} o(x', y') \delta(x' - k\cos\theta) dx' dy' = \iint_{-\infty}^{\infty} o(x', y') \delta(x' - k') dx' dy', \quad (3)$$

where $[x', y']^T = R_\theta [x, y]^T$, and R_θ is the rotation matrix. $k' = k\cos\theta$ is the resampling operation as explained in Section 1 of Supplementary Materials.”

16. Line 358: Can you comment on the necessity of the scenes to be sparse? This is clearly encoded in the choice of $\phi(g) = g$, but is not directly addressed in the main paper. However, it is clear that the light in the fiber, the NLOS scenes, and the training data for the NN inverse are all highly sparse. This helps explain why sparsity in various transform domains would not make a significant difference. How complicated can the scene get though? Would TV regularization be enough for more complex scenes that do not simply have a bright patch of interest surrounded by a black background?

Response:

Please see our response to comments 2. Also, Section 6.2 of Supplementary Materials showed synthetic result of applying LIFT (including using only 7 lenslets) for imaging complex NLOS scenes. We show below LIFT reconstruction with different transform function $\phi(g)$ for a cluttered camera-man scene. As the sentence “In our tests, these transformations yield similar results in most cases.” is on the subjective side, we deleted it in the revised manuscript.

Figure R1, LIFT reconstruction of camera-man photograph (128 x 128) using different transform functions under different number of projections.

17. Lines 366-372: This analysis appears to be only for reconstructing 2D images over time. Since each time slice is processed independently, the analysis could be shown just for a single 2D reconstruction. Is there also analysis for the light field reconstruction (from different perspectives)?

Response:

We thank the reviewer for raising this point. For light field reconstruction, LIFT needs to shear-and-reconstruct onto different depths, with each depth being independently recovered just as for each time slice. With N_d depths, the shearing step is comparatively trivial: $O(nNN_d)$, and the reconstruction complexity can be obtained by changing N_t to N_d , as each depth are independently processed, leading to $O(mnN^2N_d)$. The complexity of computing the focal measure is $O(kN^2N_d)$: focal measure for each pixel (of image at each depth) is computed around a patch of k pixel. Mapping to a depth value across the focal stack is $O(N^2N_d)$. Hence, for 3D reconstruction, the total complexity is dominated by the reconstruction step at $O(mnN^2N_d)$.

Regarding this, we added in the last paragraph of Image Reconstruction section of Methods that reads “Similarly, with a depth resolution of N_d , the reconstruction complexity for a 3D scene (x, y, z) is $O(mnN^2N_d)$: each depth is reconstructed independently after shearing the measurement data (see Section 3 of Supplementary Materials for the working flow of reconstructing 2D, 3D, and 4D images).”

Supplementary Information

18. Eqn. S5 is key to answering some of my questions about how the light field reconstruction can be performed (due to the u term) and may be helpful to include in the main text as well.

Response:

Please see our response to comment 1.

19. Fig. S2- is there a panel (g) missing?

Response:

We corrected this error by changing panel h to g in the revised Fig. S2.

20. Lines 91-92: What does it mean “Actual implementation of LIFT usually restricts the number of projections in lieu of ten”? Is it supposed to imply the number of projections is restricted to less than ten?

Response:

We intend to mean when given a limited pixel number of 1D sensors (a few thousands) and the interest of obtaining an imaging resolution more than 100×100 , the number of projections is about 10, but not restricted to less than ten. For instance, using a 1D sensor with a pixel resolution of 4096, the number of lenslet can be about 30 to achieve a LIFT image resolution of 100×100 . To clarify, we revised the sentence to read “...Using 1D sensors with a limited pixel count (several thousands) for an image resolution over 100×100 , practical implementation of LIFT usually restricts the number of projections on the order of ten.”

21. Line 140: “LIFT captures only the angular information along one axis (Q) instead of two”---what effect does this have on 3D reconstruction? The results show that depth can be reconstructed with only 1 dimension of angular information. How much would the second axis help?

Response:

For depth reconstruction, the second dimension will not help much: the base-line size is the most important parameter and with the same base-line, a stereo-camera (two views only) can achieve the same depth extraction accuracy as a light field camera with multiple views. This was analysed in the literature [1] “*Depth from defocus vs. stereo: how different really are they?*”. Also, light field camera employing the EPI (epipolar image) method for depth extraction only exploited 1D rather than 2D angular information. Adding the second dimension via camera rotation or camera array will increase the number of views (projections) and therefore enhance the 2D reconstruction quality, which ultimately improves the 3D imaging quality. Regarding this, we add a sentence to the last paragraph of Section 3.1 of Supplementary Materials that reads “A 2D angular information will benefit LIFT with an enhanced

2D reconstruction as it yields a larger number of projections and, consequently, improve 3D reconstruction as well.”

22. S3.3: It would be great to include the actual optimized lenslet angles for the depth-of-field and depth-sense versions of the lenslet arrangements for the 7-lenslet implementations.

Response:

We included the optimized lenslet arrangements for both versions in the revised Fig. 3d, as appended below.

Figure S3d, Experimental lenslet arrangement for the depth-of-field and depth-sense version of LIFT, with black solid line representing the invariant axis of the cylindrical lenslet. The angles underneath each lenslet is w.r.t the y axis (counterclockwise being the positive direction) and listed with an accuracy of 1 degree for clarity.

23. Line 213: What metric is used as a focus measure to be able to infer depth?

Response:

We implemented the sum of modified Laplacian as the focal measure, which is included in the revised Line 213 “... To infer depth, a focus **measure (sum of modified Laplacian⁶)** is computed ...”.

24. Is there any section that details the actual experimental hardware used (e.g., manufacturer, basic specifications, etc.)?

Response:

We detailed the manufacturer and basic specifications of the used hardware in Section 6.1 of the revised **Supplementary Materials** as appended below.

“6.1 *Experimental setup*. We summarize below the equipments used in LIFT system.

1. Streak camera: C13410-01A (Hamamatsu Photonics), 10000:1 dynamic imaging range, effective image resolution: 1314 (slit direction) × 1016 (time direction), frame rate: 100 Hz (storing or transferring). Observation window: variable from 500 ps to 1 millisecond long.
2. Cylindrical lenslet: plano-convex, custom-made, 2 mm diameter, 8 mm focal length.

3. Ultrafast photodiode: 818-BB-45 (Newport Inc.), 500 nm~ 890 nm, rise time ~30 ps.
4. Picosecond laser: Spark Sirius (Spark-Lasers Inc.), 532 nm, 6 ps pulse width, 2 mW average power at 100 Hz repetition rate.
5. Video camera: Hero4 Silver (GoPro Inc.), 1080p at 60 Hz maximum.”

References

1. Schechner, Y. Y. & Kiryati, N. Depth from defocus vs. stereo: how different really are they? in *Proceedings. Fourteenth International Conference on Pattern Recognition (Cat. No.98EX170)* vol. 2 1784–1786 (IEEE Comput. Soc, 1998).

REVIEWERS' COMMENTS

Reviewer #2 (Remarks to the Author):

The authors have addressed my comments. I recommend its publication in NC.

Reviewer #3 (Remarks to the Author):

I continue to be impressed by the innovative approach to imaging presented by this manuscript, and I look forward to seeing how light-field tomography can be extended to other applications. Overall, the changes made by the authors have addressed my concerns and improved the manuscript.

My only remaining comment is for Figure S3: Are all the lenslet angles listed correctly? In the depth-of-field arrangement, it looks like -22, -37 should be positive valued.

Response Letter

Reviewer 1

The authors have addressed my comments. I recommend its publication in NC.

Response:

We thank the reviewer for the positive comments on this work.

Reviewer 2

I continue to be impressed by the innovative approach to imaging presented by this manuscript, and I look forward to seeing how light-field tomography can be extended to other applications.

Overall, the changes made by the authors have addressed my concerns and improved the manuscript.

My only remaining comment is for Figure S3: Are all the lenslet angles listed correctly? In the depth-of-field arrangement, it looks like -22, -37 should be positive valued.

Response:

We thank the reviewer for appreciating our work and the typos were corrected in the revised Figure S3.